DISCOVERY REPORT

# Microbe transmission from pet shop to lab-reared zebrafish reveals a pathogenic birnavirus

**Marlen C. Rice**[1], **Andrew J. Janik**[1,2], **Nels C. Elde**[2,3], **James A. Gagnon**[1]*, **Keir M. Balla**[2¤]*

**1** School of Biological Sciences, University of Utah, Salt Lake City, Utah, United States of America, **2** Department of Human Genetics, University of Utah School of Medicine, Salt Lake City, Utah, United States of America, **3** Howard Hughes Medical Institute, Chevy Chase, Maryland, United States of America

¤ Current address: Chan Zuckerberg Biohub, San Francisco, California, United States of America
* james.gagnon@utah.edu (JAG); keir.balla@czbiohub.org (KMB)

The Editors encourage authors to publish research updates to this article type. Please follow the link in the citation below to view any related articles.

**Data Availability Statement:** Data are available within the paper and its Supporting Information

## Abstract

Zebrafish are popular research organisms selected for laboratory use due in part to widespread availability from the pet trade. Many contemporary colonies of laboratory zebrafish are maintained in aquaculture facilities that monitor and aim to curb infections that can negatively affect colony health and confound experiments. The impact of laboratory control on the microbial constituents associated with zebrafish in research environments compared to the pet trade are unclear. Diseases of unknown causes are common in both environments. We conducted a metatranscriptomic survey to broadly compare the zebrafish-associated microbes in pet trade and laboratory environments. We detected many microbes in animals from the pet trade that were not found in laboratory animals. Cohousing experiments revealed several transmissible microbes including a newly described non-enveloped, double-stranded RNA virus in the Birnaviridae family we name Rocky Mountain birnavirus (RMBV). Infections were detected in asymptomatic animals from the pet trade, but when transmitted to laboratory animals RMBV was associated with pronounced antiviral responses and hemorrhagic disease. These experiments highlight the pet trade as a distinct source of diverse microbes that associate with zebrafish and establish a paradigm for the discovery of newly described pathogenic viruses and other infectious microbes that can be developed for study in the laboratory.

## Introduction

The zebrafish is one of the most common fish maintained in aquaria and has been propagated in the pet trade since at least the early 20th century [1]. Accessibility is one of the driving factors that originally led biologists to adopt zebrafish as research organisms in laboratories [1–3], and most of the laboratory lines in use today were originally derived from pet trade sources [4]. Many contemporary populations of laboratory zebrafish have been propagated in controlled housing systems for several decades under supervisory regimes that typically aim to minimize the abundance of potentially pathogenic microbes that are prevalent in pet trade

files, and raw sequencing files are available at NCBI under BioProject PRJNA1005695.

**Funding:** This work was supported by grants awarded by the National Institutes of Health to K.M. B. (5T32AI055434), N.C.E. (R35GM134936), and to J.A.G. (R35GM142950), and by a grant from the Chan Zuckerberg Initiative to N.C.E. and J.A.G. (DAF2020-218441). The funders played no role in the study design, data collection and analysis, decision to publish, or preparation of the manuscript.

**Competing interests:** The authors have declared that no competing interests exist.

**Abbreviations:** IPNV, infectious pancreatic necrosis virus; RdRp, RNA-dependent RNA polymerase; RMBV, Rocky Mountain birnavirus; SVCV, spring viremia of carp virus; ZfPV, zebrafish picornavirus.

environments [5–7]. The long-term investments in maintaining standardized research organisms might profoundly alter the types and abundances of microbes that associate with zebrafish in the laboratory relative to pet trade sources. Complicating matters, zebrafish are still occasionally imported from the pet trade into laboratory colonies [7,8] and our knowledge of the zebrafish-associated microbes in pet trade and laboratory environments is limited.

Increasing our knowledge of the microbes that associate with zebrafish in different environments could reveal chains of microbial transmission, improve research colony health monitoring programs, and lead to the discovery of unknown infectious microbes. These studies have the potential for establishing new experimental systems for investigating infection biology with broad biomedical and aquaculture relevance. The current set of known zebrafish-associated microbes that are commonly found in laboratory environments includes stable communities of bacteria that comprise the zebrafish microbiota [9,10], bacterial pathogens that cause mycobacteriosis [11], parasitic nematodes and other eukaryotic pathogens such as microsporidia [12,13], and a picornavirus distantly related to poliovirus [14,15]. Few studies have investigated the zebrafish-associated microbes in pet trade environments, but recently discovered pathogens in class Trematoda and Conoidasida in pet trade zebrafish [7,8] highlight these environments as rich sources of newly observed microbes. Histopathology and 16S rRNA sequencing are the primary tools that have been used in the detection and discovery of zebrafish-associated microbes in laboratory and pet trade environments. Identifying newly described viruses with these methods is often impossible, and other classes of microbes can also be challenging to detect. Here, we use metatranscriptomic sequencing and cohousing experiments to survey the microbes that can inhabit and transmit between zebrafish populations in the laboratory and pet trade. We report the newly described Rocky Mountain birnavirus (RMBV) from asymptomatic pet trade zebrafish that appears highly transmissible and pathogenic in laboratory-reared zebrafish.

## Results

### Zebrafish from the pet trade harbor known and novel microbes that are potentially pathogenic

We sampled outwardly healthy zebrafish from 3 different pet trade sources and a research laboratory housing system to survey microbial populations across environments through sequencing of bulk intestinal RNA. After excluding reads that mapped to the *Danio rerio* genome, we performed de novo assembly of transcripts and assigned prospective taxonomic classification based on nucleotide or amino acid similarity to known sequences (S1 and S2 Files).

We classified hundreds of zebrafish-associated transcripts as viral and thousands of transcripts as prokaryotic or eukaryotic (S2 File). Many prospective virus sequences were derived from bacteriophages, while others had similarity to viruses that infect diverse eukaryotic hosts. Among the virus families that infect vertebrate hosts, we reconstructed 2 transcripts that constitute a full genome sequence (GenBank accession numbers OR427289 and OR427290) for a novel bisegmented double-stranded RNA virus with similarity to birnaviruses that infect fish (Fig 1A). Phylogenetic analyses confirmed that the birnavirus we discovered, which we named RMBV, is part of a clade of viruses that infect fish and other vertebrates within the family Birnaviridae (Fig 1B). This clade includes infectious pancreatic necrosis virus (IPNV), which can cause severe disease and losses in aquaculture. The closest relative to RMBV in this phylogeny, Wenling jack mackerels birnavirus, shares 62% amino acid identity in the RdRp protein. Thus, RMBV is a newly described virus that appears to naturally infect zebrafish and is related to significant pathogens in aquaculture.

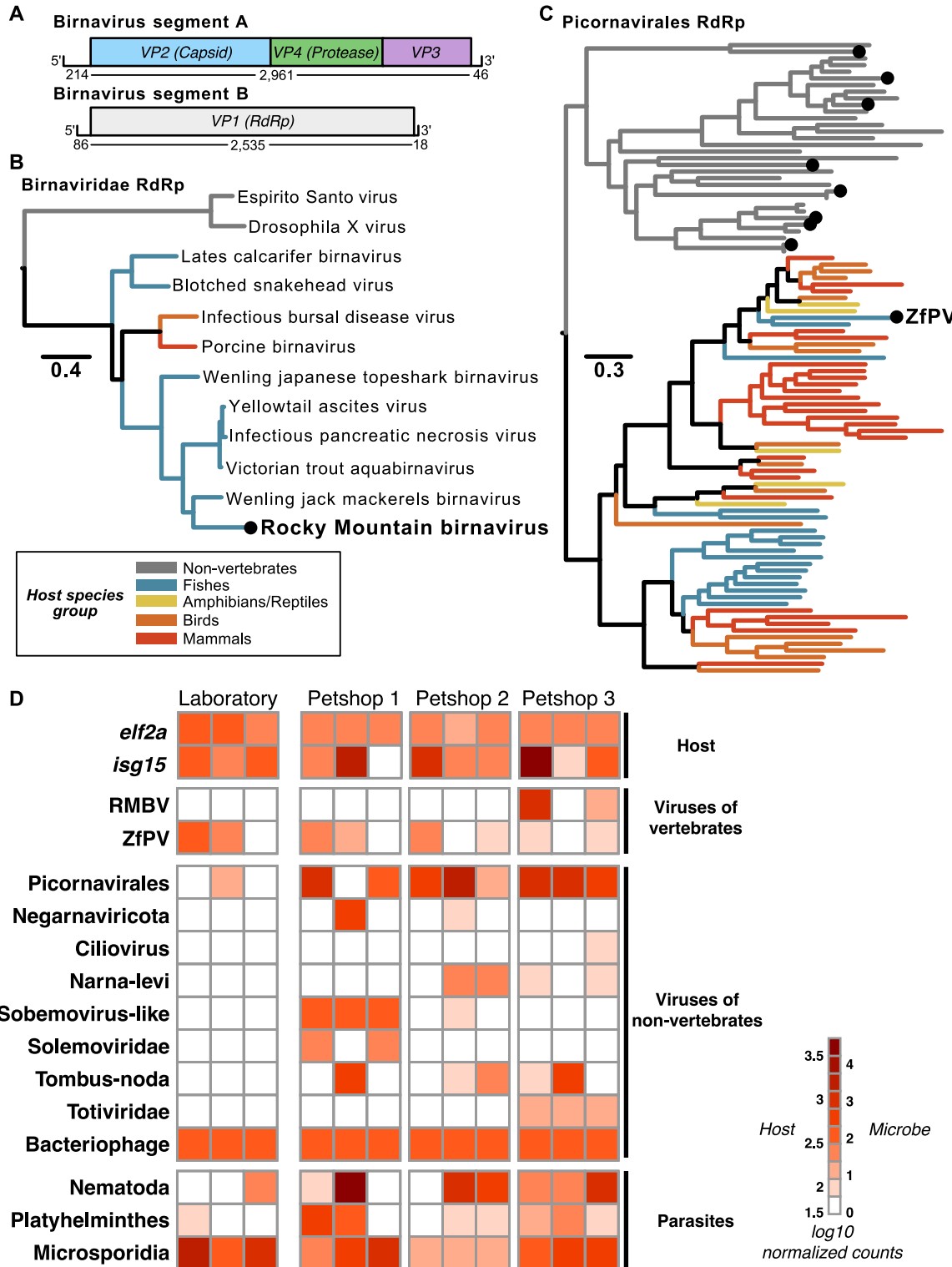

**Fig 1. Metatranscriptomic survey of microbes associated with zebrafish from laboratory and pet trade sources.** . (A) Schematic representation of a newly discovered birnavirus associated with zebrafish in this study. Both segments of the linear dsRNA genome are depicted with boxes denoting relative positions of ORFs based on protein domain conservation. Numbers beneath segments indicate nucleotide lengths of each prospective 5′ UTR, coding sequence, and 3′ UTR. (B) Maximum likelihood phylogeny of viruses in the family Birnaviridae. The black dot highlights the novel zebrafish-associated virus schematized in (A), which we name RMBV. (C) Maximum

likelihood phylogeny of RdRp protein sequences from viruses in the order Picornavirales. Black dots highlight the zebrafish-associated viruses that were detected (ZfPV) or newly discovered in this study. Trees in (A and C) are midpoint rooted. Scales are in amino acid substitutions per site. Branches are colored based on the known or inferred host group. (D) Taxonomic classification and quantification of host and microbe transcripts identified in intestinal tissues of zebrafish from laboratory and pet trade sources. Columns show estimated abundances in tissue samples from independent individuals grouped by source. RdRp, RNA-dependent RNA polymerase; RMBV, Rocky Mountain birnavirus; ZfPV, zebrafish picornavirus.

Among the other viruses with known or possible vertebrate hosts, we detected an endemic zebrafish picornavirus (ZfPV) that we and others recently discovered [14,15] and dozens of other sequences with similarity to members of the order Picornavirales. Eight of the novel sequences contained full-length RNA-dependent RNA polymerase (RdRp) sequences that we used along with representative RdRp sequences across the order Picornavirales to perform phylogenetic analyses. A maximum likelihood phylogeny placed all novel picornavirus-like sequences in a clade with viruses isolated from non-vertebrate hosts (Fig 1C), which suggests that they infect eukaryotes associated with zebrafish as opposed to zebrafish themselves.

We next estimated host and microbe abundances by mapping all reads to an integrated set of zebrafish and microbial transcripts (Fig 1D and S3 File). ZfPV was broadly detected in zebrafish from both laboratory and pet trade sources, extending its known presence beyond the laboratory to other environments [14,15]. In contrast, RMBV was only detected in zebrafish from a single pet trade source. Zebrafish from this source also expressed significantly higher levels of interferon-stimulated genes than control zebrafish from the laboratory, while zebrafish from the other pet trade sources did not exhibit signatures of antiviral gene expression (S1 Fig and S4 File). These comparisons reveal a canonical antiviral defense response associated with the presence of RMBV in the absence of visible disease phenotypes in pet trade animals.

Most of the other prospective eukaryote-infecting viruses we discovered were only detected in pet trade sources. The closest match to a novel virus in phylum Negarnaviricota discovered here is Amsterdam virus (61% amino acid identity), which is associated with parasitic nematodes that infect rodents [16]. Together with a pronounced abundance of nematode transcripts in the pet trade zebrafish, these observations suggest that the newly described negative-sense RNA virus infected nematodes that infected zebrafish. Parasitic worm transcripts were detected in animals from both environments but were generally more abundant in zebrafish from the pet trade, while microsporidia transcripts were abundant in both laboratory and pet trade zebrafish. Clustering samples based on the abundance of bacteria families in our data did not distinguish between their sources (S5 File), highlighting that similar bacterial communities were associated with zebrafish in laboratory and pet trade environments. Thus, we did not observe identifiable bacterial infections specific to pet trade zebrafish. The results from this metatranscriptomic survey underscore the use of zebrafish from the pet trade as potential sources of viruses and eukaryotic parasites that are not commonly found in laboratory zebrafish.

## Transmission of RMBV from pet trade to laboratory zebrafish is associated with necrotic disease

To determine if microbes found in pet trade zebrafish could be naturally transmitted to laboratory zebrafish, we devised a cohousing experiment using closed circulation aquaria containing zebrafish from both environments (Fig 2A). A control tank was seeded with only zebrafish from the laboratory, and a cohousing tank was seeded with zebrafish from the laboratory along with zebrafish from pet trade source 3 (Fig 1D). All zebrafish in the control tank remained healthy throughout the experiment (Fig 2B). In contrast, we observed hemorrhaging

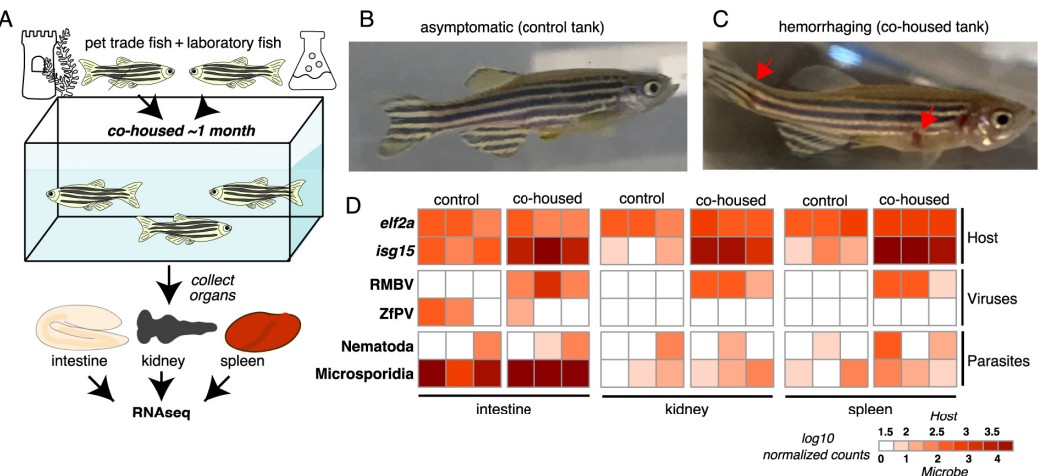

**Fig 2. Overt disease observed in laboratory zebrafish cohoused with zebrafish from the pet trade is associated with RMBV infection.** (A) Diagram of cohousing experiment. Tanks were seeded with adult zebrafish from the laboratory (control) or zebrafish from the laboratory and the pet trade (cohoused). (B) A representative individual from the control tank. (C) A representative laboratory zebrafish 1 month after cohousing with zebrafish from the pet trade. Red arrows highlight necrotic lesions. (D) Transmission of microbes from pet trade to cohoused laboratory zebrafish inferred from host and microbe transcript abundances. Three laboratory fish were sampled from each tank. Samples are grouped by tank and tissue. Columns show estimated transcript abundances in tissue samples from individual laboratory zebrafish. RMBV, Rocky Mountain birnavirus.

in 3 laboratory zebrafish approximately 1 month after cohousing them with pet trade zebrafish (Fig 2C). At this point, we collected intestines, kidneys, and spleens from control and cohoused laboratory zebrafish for RNA isolation and sequencing. The metatranscriptomes of these samples revealed pronounced levels of RMBV in the 3 diseased laboratory zebrafish that were cohoused with pet trade zebrafish (Fig 2D). We detected RMBV in all tissues that were sampled, revealing that infections were systemic. No other viruses that were unique to pet trade zebrafish in our survey above (Fig 1D) were detected in cohoused laboratory zebrafish. RNA was not collected from the 7 cohoused zebrafish that did not exhibit disease, and the infection status of these animals is unknown. These experiments demonstrate that viruses circulating in seemingly healthy zebrafish from the pet trade can be naturally transmitted to laboratory zebrafish and cause disease.

In a second cohousing experiment, another tank was seeded with zebrafish from the laboratory along with additional zebrafish from pet trade source 3. Zebrafish were cohoused for the same length of time as in the experiment described above but no hemorrhaging was observed in laboratory or pet trade zebrafish. We collected intestines from pet trade and laboratory zebrafish after 1 month of cohousing for RNA isolation and sequencing, which revealed RMBV transcripts in one of 2 pet trade zebrafish but no evidence of RMBV transcripts in either of the 2 cohoused laboratory zebrafish (S2 Fig and S6 File). These results demonstrate that in some contexts, zebrafish can carry RMBV without transmitting infection to potential recipients in a shared environment. Although we detected no evidence for the transmission of RMBV in this experiment, we observed exceptionally high levels of parasitic nematode transcripts in all cohoused zebrafish. Furthermore, we detected transcripts from the novel virus in phylum Negarnaviricota described above in cohoused pet trade and laboratory zebrafish, lending additional support to the proposition that nematodes serve as hosts. These experiments highlight examples of virus and parasite transmission from pet trade to cohoused laboratory zebrafish and reveal variability in the frequency of transmission.

## Naturally transmitted RMBV infections elicit inflammatory immune responses in laboratory zebrafish

To identify host transcriptional responses to RMBV, we tested for differential gene expression in tissues from infected laboratory zebrafish compared to control laboratory zebrafish and identified more than 800 genes at an adjusted $p < 0.05$ that were induced by at least 2-fold (S7 File). RMBV infection was associated with elevated levels of genes related to interferon signaling, inflammation, innate immunity, and adaptive immunity (Fig 3A). We identified an overlapping set of differentially expressed genes in all 3 organs, but the spleen had more virus-induced genes than infected intestine or kidney tissues (Fig 3B and S7 File). We next compared the RMBV-induced genes to those induced in another virus infection context. Spring viremia of carp virus (SVCV) is an RNA virus naturally found in a range of fish species that can also cause hemorrhagic disease when experimentally administered to zebrafish [17]. A recent study measured gene expression in zebrafish intestine, kidney, and spleen tissues 12 h after intraperitoneal injection with SVCV [18]. Despite several differences in experimental context, more than a third of the genes that were highly induced by RMBV were also induced during infection with SVCV (Fig 3C). We observed enrichment for antiviral defense processes among both virus-induced gene sets, enrichment for B cell activation among RMBV-induced genes, and enrichment for lymphocyte chemotaxis and other processes among SVCV-induced genes (Fig 3D). We speculate that the divergent enrichment for lymphocyte activation and chemotaxis between RMBV- and SVCV-induced gene sets may reflect differences in the stage of infection progression, but defining the full extent of overlap between the gene sets induced by either virus will require further experimentation. These observations highlight genes associated with inflammatory disease observed during natural transmission of birnavirus infections in zebrafish.

## Discussion

This study presents an experimental platform for unbiased molecular detection and discovery of microbes circulating in zebrafish from different environments. We used this integrated workflow to discover a newly described naturally transmissible birnavirus that is associated with hemorrhagic symptoms in some infected individuals. Phylogenetic analyses place RMBV in a clade with infectious pancreatic necrosis virus, which is among the most significant threats to global aquaculture industry due to the fatal hemorrhagic disease that can result from infection [19]. Survivors of IPNV infection can become lifelong carriers that transmit virus both horizontally and vertically [20]. These observations motivated the first study to use zebrafish for investigating virus infection biology, in which the authors demonstrated that IPNV can transmit to the eggs of infected adults and persist in the next generation [21]. Subsequent studies have confirmed that IPNV can infect zebrafish but does not cause observable disease [22,23]. In contrast, we observed clear signs of hemorrhaging associated with RMBV infection in zebrafish, suggesting that this virus–host pair might better recapitulate features of disease caused by birnaviruses in aquaculture settings. We observed transmission of RMBV infection between zebrafish in a shared environment through an unknown route. Future investigations of horizontal transmission and the possibility for vertical transmission will be important for establishing measures that might be implemented to control the spread of RMBV infections in zebrafish colonies.

Unlike the laboratory-reared RMBV-infected zebrafish, infected zebrafish from the pet trade did not exhibit signs of disease. Studies that cohoused mice from the laboratory and the pet trade also observed disease that was restricted to the laboratory-reared cohort [24,25], which was linked to a relative deficit of microbial exposure and immune memory in laboratory

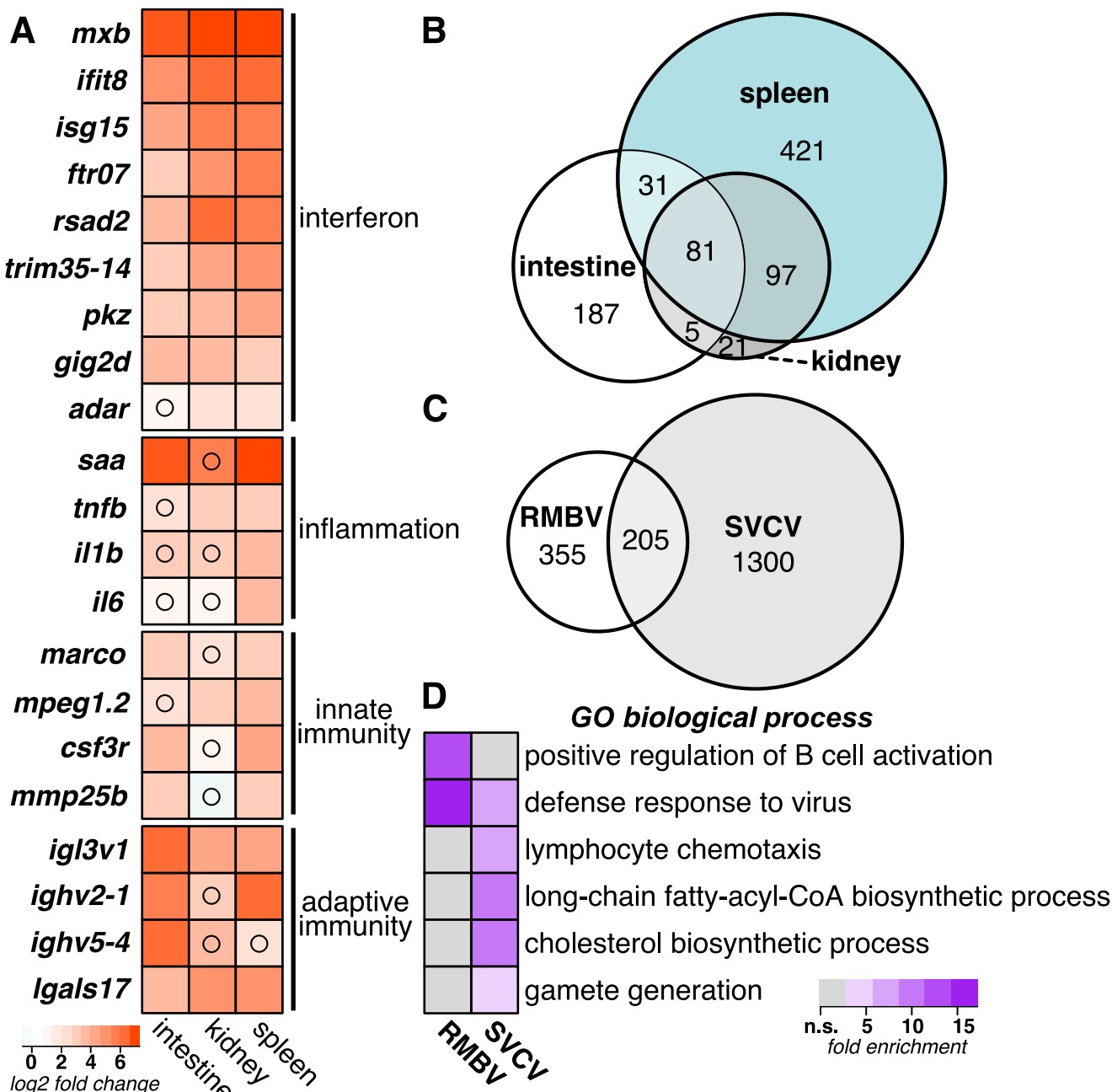

**Fig 3. Naturally transmitted RMBV infections induce systemic inflammatory antiviral responses in laboratory zebrafish.** (A) Differential gene expression in laboratory zebrafish infected by RMBV compared to uninfected laboratory zebrafish. Average log2 fold change expression is plotted for each gene in a comparison of 3 RMBV-infected and 3 uninfected samples for each tissue. All comparisons had adjusted *p*-values of <0.05 unless denoted with a black circle. (B) Comparison of RMBV-induced gene sets by tissue. Euler plot includes all genes with log2 fold changes >1 and adjusted *p*-values <0.05. (C) Comparison of virus-induced gene sets in intestine, spleen, or kidney during RMBV or SVCV infections. Euler plot includes all genes with log2 fold changes >2 and adjusted *p*-values <0.05. SVCV RNAseq data are from (PMID: 34908463). (D) Enrichment of Gene Ontology biological processes among the genes that are induced with log2 fold changes >2 during RMBV or SVCV infections. All biological processes with false discovery rates of <0.05 for at least 1 condition are plotted. RMBV, Rocky Mountain birnavirus; SVCV, spring viremia of carp virus.

mice compared to mice from the pet trade or the wild [24]. The differences in disease manifestation that we observed between laboratory-reared and pet trade zebrafish infected by RMBV could result from differences in immunological life history, genetics, stage of infection, or a combination of influences. The antiviral transcriptional signatures that we observed in pet trade zebrafish infected by RMBV demonstrate that transcriptional status is a consistent phenotype of infection regardless of visible disease. Identifying the causes of variation in birnavirus virulence is an ongoing effort in research with fish [19], birds [26], and mammals [27]. Future work aimed at isolating or synthesizing infectious RMBV would enable new avenues of research using zebrafish to resolve determinants of birnavirus infection, disease, transmission, and immunity in vertebrates.

The outcome of any infection is largely dictated by the specific pairing of host and virus genomes. IPNV was first isolated from brook trout [28] but has since been isolated from various species of farmed and wild fish around the world [29]. Disease resulting from IPNV infection in salmonids is often associated with transcriptional activation of interferon signaling, genes involved in inflammation, and the adaptive immune system [30–34]. Experimental administration of IPNV in zebrafish leads to infection without induction of interferon signaling or disease [22,23]. We observed that naturally occurring RMBV infections in laboratory-reared zebrafish are associated with inflammatory antiviral gene expression programs and, at least in some animals, clear signs of necrotic disease. Spring viremia of carp virus infections in zebrafish also lead to necrotic disease [17] associated with transcriptional activation of interferon signaling and inflammation [18,35,36]. Extending these comparisons to comprehensive assessment of the transcriptional programs induced by RMBV, SVCV, and other viruses associated with inflammation along with comparisons to the gene sets induced by IPNV and other viruses that are not associated with inflammation in zebrafish will improve our understanding of why some infections result in disease while others do not.

In addition to RMBV, our metatranscriptomic sampling of zebrafish-associated microbes provides a suite of molecular identifiers for known and newly described members of communities in laboratory and pet trade environments. The large diversity of sequences from cellular organisms included in this set makes specific host assignment challenging for some of the newly described viruses. However, phylogenetic analyses strongly support a non-vertebrate host for most virus sequences we detected. While it is not yet clear how the variety of viruses infecting non-vertebrate hosts might affect zebrafish, some of the viruses appear to hold that potential. One of the newly described negative-strand RNA viruses in our dataset is most similar to a virus associated with parasitic nematode infections in mice [16], which suggests that a nematode that infects zebrafish is the natural host. Viruses of parasites are potentially meaningful as they can alter the virulence associated with parasite–host interactions [37]. Nematode sequences were among the most abundant in our dataset, which is consistent with the known prevalence of these parasites in zebrafish in laboratory and pet trade environments [38]. While nematode infections in zebrafish are well-documented by histology, our study expands the repertoire of nematode sequences that can be used to assess the health of zebrafish colonies.

We also identified hundreds of transcripts from at least 1 newly described species of microsporidia. Whole genome data exist for one of the 2 species of microsporidia that have been described in zebrafish [39]. The microsporidia sequences in our dataset may therefore derive from a parasite previously described by histological studies or from a newly described species.

Altogether, the diversity of sequences we uncovered in a sparse sampling of zebrafish from laboratory and pet trade environments suggests that additional sampling will reveal many more newly described microbes. These discoveries will expand our ability to monitor the health of zebrafish colonies and lead to new experimental opportunities for studying infection biology with a versatile research organism.

## Materials and methods

### Pet trade survey and cohousing assay

This study was conducted under the approval of the Office of Institutional Animal Care and Use Committee (IACUC # 00001439) of the University of Utah's animal care and use program. Nine zebrafish from 3 local pet trade sources in the Salt Lake Valley were purchased. Fish were euthanized and intestinal tissue harvested and processed individually for RNA extraction. Cohousing assay was set up with closed circulation aquaria using 3 identical 10-gallon tanks. Tanks were inoculated with ceramic filter rings (Aquapapa), these filters had been housed in a zebrafish research facility for 1 month previously to build up a nitrifying microbial community. Identical filtration systems were set up in each tank with activated charcoal and sponge filters (AquaClear 50). Tanks were cycled for 6 weeks using pure ammonia (Great Value). The cohousing experiments were initiated when ammonia and nitrite levels were undetectable. Each tank housed 15 fish. Fish were fed identical diets (Zeigler, Adult Zebrafish Diet). Control tanks contained 15 laboratory-reared Tübingen zebrafish. The laboratory samples presented in Fig 1 were collected from the control tank of zebrafish. Each cohousing tank contained 10 laboratory-reared Tübingen fish and 5 pet trade fish. Pet trade animals were obtained from the same source for both cohousing experiments and their genetic backgrounds were unknown. In the first experiment, the pet trade fish were all females and the laboratory fish were all males (Fig 2), and in the second experiment, the pet trade fish were all males and the laboratory fish were all females (S2 Fig). No fighting or evidence of aggression was observed throughout the cohousing experiments. None of the control or cohoused fish exhibited abnormal swimming behaviors throughout the experiments. Approximately 1 month into the first cohousing experiment several laboratory-reared fish began exhibiting hemorrhaging. These were euthanized and processed for RNA extraction. All additional fish used in the study were subsequently euthanized and processed within 1 week.

### RNA extraction and sequencing

Fish were euthanized and intestines, whole kidney marrows, and spleens were harvested for processing. All tissues were homogenized using mechanical lysis. RNA was extracted using the directzol kit (Zymo), and 1 ml of TRIzol was used per intestine and 0.5 ml of TRIzol used for individual spleens and kidneys. Following RNA extraction, samples were DNAse treated (Zymo). RNA ScreenTape (Agilent) was used to assess quality of RNA samples. Only samples with RIN scores >8 were used for analysis. RNA libraries were prepared by the High Throughput Genomics Shared Resource at the University of Utah with the Illumina TruSeq Stranded Total RNA Library Prep Ribo-Zero Gold and sequenced on a Novaseq with using a $150 \times 150$ bp sequencing kit to a depth of 25 million reads per sample. Raw sequencing reads were deposited at NCBI under BioProject PRJNA1005695.

### Metatranscriptomic classification and quantification of RNA-sequencing data

Reads were mapped to the GRCz11 Danio rerio genome assembly (Ensembl release 100) with STAR v2.7.3a [40]. Unmapped reads from all samples were concatenated and assembled de novo using Trinity v2.11.0 [41]. Prospective taxonomic lineages were assigned to each assembled Trinity "gene" by extracting the best hit (highest bit score) per query from BLASTn [42] or DIAMOND v2.0.14 [43] searches of the nt or nr NCBI sequence collections (downloaded in May 2022), respectively. All contigs prospectively assigned to microbial lineages were then aligned against all available zebrafish nucleotide sequences with BLASTn. Any contig that

aligned a zebrafish sequence with a bitscore of 100 or greater was removed from the collection of contigs. Estimated zebrafish and microbe transcript counts were obtained by mapping reads from each sample to an index of zebrafish transcripts and all newly assembled sequences using Salmon v1.3.0 [44] with the validateMappings flag and 20 Gibbs samples. Counts were summed at the gene or taxonomic family level, and normalized counts were generated using DESeq2 v.28.1 [45].

## Phylogenetic analyses

Phylogenies were estimated for viruses in the order Picornavirales using amino acid sequences from prospective 3C protease and 3D RNA-dependent RNA polymerase genes. Sequences from at least 1 species of all known genera in family Picornaviridae were included, as well as all available sequences from picornaviruses that infect fish. Several representative viruses from other clades were also included to span the currently known breadth of Picornavirales. Sequences were aligned with MAFFT version 7 [46] and manually trimmed to remove sequences at ends with mostly empty columns, resulting in an alignment of 753 amino acid sites across 95 taxa. Evolutionary relationships were inferred using maximum likelihood phylogenetic analyses carried out in RAxML-NG version 1.0.2 [47]. The LG+I+G+F substitution model was selected as the best-fit model for the alignment under the Akaike information criterion as determined by ProtTest version 3.4.2 [48]. Bootstrapping converged after 650 replicates. Birnavirus phylogenies were estimated using amino acid sequences from VP1 (RdRp) genes. Representative viruses were included from vertebrate and invertebrate hosts. Sequences were aligned using MAFFT-L-INS-i with DASH structures, resulting in an alignment of 1,093 amino acid sites across 12 taxa. Evolutionary relationships were inferred using maximum likelihood phylogenetic analyses carried out in RAxML-NG [47]. The LG+G+I+F substitution model was selected as the best-fit model for the alignment under the Akaike information criterion as determined by ProtTest [48]. Bootstrapping converged after 300 replicates.

## Comparisons to Spring viremia of carp virus

Raw fastq reads in BioProject PRJNA690234 [18] were downloaded from the Sequence Read Archive at NCBI with the SRA Toolkit. Estimated zebrafish and microbe transcript counts were obtained by mapping reads from each sample to the same index of zebrafish transcripts and all newly assembled sequences using Salmon v1.3.0 [44] as described above. Counts were summed at the gene or taxonomic family level, and normalized counts were generated using DESeq2 v.28.1 [45]. All genes with log2-fold changes greater than 2 and adjusted $p$-values less than 0.05 were included in comparisons of birnavirus- and SVCV-induced gene sets. Euler plots were generated in R with the eulerr package [49]. Gene Ontology enrichment was determined with the PANTHER overrepresentation test (released February 2022) [50].

## Supporting information

**S1 Fig. Antiviral gene expression in zebrafish from pet trade sources compared to control zebrafish from the laboratory.** Differential gene expression was measured for the 3 replicates from each pet trade source compared to the 3 laboratory replicates (data in S4 File). Log2 fold change values are plotted for a selected set of antiviral genes with adjusted $p$-values <0.1. (PDF)

**S2 Fig. Zebrafish from the pet trade can harbor RMBV without transmitting to cohoused zebrafish from the laboratory.** (A) Diagram of cohousing experiment. Tanks were seeded with adult zebrafish from the laboratory and pet trade source 3, as in Fig 2. (B) Transmission

of microbes from pet trade to cohoused laboratory zebrafish inferred from host and microbe transcript abundances. Intestines from 2 donor pet trade zebrafish and 2 laboratory recipient fish were sampled after 1 month of cohousing. Control zebrafish are the same as shown in Fig 2. Columns show estimated transcript abundances in intestine samples from individual zebrafish.
(PDF)

**S1 File. Transcripts assembled from bulk RNA-sequencing data of adult zebrafish intestines from the laboratory and 3 pet trade sources.**
(TXT)

**S2 File. Taxonomic lineage assignments for all assembled transcripts.**
(CSV)

**S3 File. Normalized counts for all microbe families and zebrafish genes for all RNA-sequencing samples included in this study.**
(CSV)

**S4 File. Differential gene expression analyses of intestine samples from each of the pet trade sources compared to the laboratory intestine samples.**
(XLSX)

**S5 File. Gut microbiome composition analyses of zebrafish from laboratory and pet trade sources.**
(PDF)

**S6 File. Log10 normalized counts of zebrafish genes and microbe transcripts plotted in S2 Fig.**
(CSV)

**S7 File. Differential expression analyses of tissues from laboratory animals comparing RMBV-infected to uninfected samples.**
(CSV)

## Acknowledgments

We thank the University of Utah High-Throughput Genomics Core, the Center for High Performance Computing, and the Centralized Zebrafish Animal Resource for their support.

## Author Contributions

**Conceptualization:** Marlen C. Rice, Nels C. Elde, James A. Gagnon, Keir M. Balla.

**Formal analysis:** Keir M. Balla.

**Funding acquisition:** Nels C. Elde, James A. Gagnon, Keir M. Balla.

**Investigation:** Marlen C. Rice, Keir M. Balla.

**Methodology:** Marlen C. Rice, Keir M. Balla.

**Resources:** Andrew J. Janik.

**Software:** Keir M. Balla.

**Supervision:** James A. Gagnon, Keir M. Balla.

**Visualization:** Marlen C. Rice, James A. Gagnon, Keir M. Balla.

**Writing – original draft:** Marlen C. Rice, Keir M. Balla.

**Writing – review & editing:** Marlen C. Rice, Nels C. Elde, James A. Gagnon, Keir M. Balla.

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
