## [Editor Report · Decision Letter 0]

4 Oct 2023

Dear Dr. Balla, 

Thank you for submitting your manuscript entitled "Microbe transmission from pet shop to lab-reared zebrafish reveals a pathogenic birnavirus" for consideration as a Short Reports by PLOS Biology.

Your manuscript has now been evaluated by the PLOS Biology editorial staff, as well as by an academic editor with relevant expertise, and I am writing to let you know that we would like to send your submission out for external peer review.

Once your full submission is complete, your paper will undergo a series of checks in preparation for peer review. After your manuscript has passed the checks it will be sent out for review. To provide the metadata for your submission, please Login to Editorial Manager (https://www.editorialmanager.com/pbiology) within two working days, i.e. by Oct 06 2023 11:59PM.

Kind regards,

Paula

---

Senior Editor

PLOS Biology

---

## [Decision Letter · Decision Letter 1]

17 Nov 2023

Dear Dr. Balla,

Thank you for your patience while your manuscript "Microbe transmission from pet shop to lab-reared zebrafish reveals a pathogenic birnavirus" was peer-reviewed at PLOS Biology. It has now been evaluated by the PLOS Biology editors, an Academic Editor with relevant expertise, and by several independent reviewers. 

In light of the reviews, which you will find at the end of this email, we would like to invite you to revise the work to thoroughly address the reviewers' reports.

As you will see below, the reviewers agree that your work is interesting, but they all raise some questions that will need to be addressed. In particular, we think it is important that you address the issues concerning causality of disease, transmissibility, and the extent of asymptomatic infections.

Given the extent of revision needed, we cannot make a decision about publication until we have seen the revised manuscript and your response to the reviewers' comments. Your revised manuscript is likely to be sent for further evaluation by all or a subset of the reviewers.

**IMPORTANT - SUBMITTING YOUR REVISION**

*Re-submission Checklist*

*Published Peer Review*

*PLOS Data Policy*

*Blot and Gel Data Policy*

Sincerely,

Paula

---

Senior Editor

PLOS Biology

REVIEWS:

Reviewer #1: Zebrafish immunity.

Reviewer #2: Zebrafish infection.

Reviewer #3: Zebrafish infection.

Reviewer #1: This is a unique contribution from a team with a track record of discovery innovating zebrafish-virus infection models. In this case authors use metagenomic sequencing and co-housing experiments to discover novel zebrafish-associated microbes and virus infections. The authors report Rocky Mountain birnavirus (RBMV) as a natural zebrafish pathogen, and that lab animals develop disease symptoms when co-housed with infected animals from a pet shop. In the same way that zebrafish infection using the natural fish pathogen Mycobacterium marinum has evolved to become a paradigm for studying human tuberculosis and natural host-pathogen interactions, the research avenue presented here looking for natural virus infections of zebrafish can similarly inspire. 

Many of my comments are outside the scope of this first discovery report, but interesting to consider for this or future work. 

To identify RBMV, why was sequencing of intestinal RNA first performed? Can infection biology be used to support inferences from omics or co-housing experiments (where evidence can be indirect). Is isolating RBMV from infected zebrafish not possible at this stage? If possible, it would be valuable to test Koch's postulates eg. isolate virus from infected zebrafish and infect lab animal, in this way directly asking if RBMV is causing disease.

Why are pet shop zebrafish resistant to hemorrhaging but lab zebrafish are not? What if infected zebrafish are co-housed with other uninfected pet shop zebrafish (eg. from sources where RBMV was not identified)? I understand that experiments involving adult fish are not trivial, but would appreciate some perspectives on this.

3/15 lab zebrafish show signs of infection from the co-housing experiments? If 12/15 lab zebrafish do not become infected, should the authors consider to modify their conclusions?

If pet shop fish are not hemorrhaging, how can authors know that pet shop fish are infected? Is infection strictly assessed by clear signs of hemorrhaging? What is causing hemorrhaging? Can non-hemorrhaging lab zebrafish (as in the case of pet shop) also have virus? Can the authors clarify whether they used any other criteria / measurement beyond haemorrhaging to assess zebrafish health eg. some scoring system which also looked at behaviour / swimming / more general health?

How is virus transmitted? Is it fish-fish contact? Is there vertical transmission (at least in the pet shop colonies)? Can it be treated?

Reviewer #2: This manuscript describes a broad, sequencing-based approach to identifying new zebrafish pathogens taking advantage of the microbial (and pathogen) diversity of pet store obtained zebrafish. By sequencing, they are able to identify potential viral, bacterial and eukaryotic pathogens, using sequence similarity to guide them to potential pathogens of interest. Using this approach, the authors are able to characterize and reconstruct the genome of a novel birnavirus that the authors named Rocky Mountain birnavirus (RMBV). They demonstrate transmission of RMBV to naïve laboratory populations, resulting in hemorrhaging of infected populations and also investigate the inflammatory changes of infection. Overall, these experiments are well done and establish an interesting pipeline for the discovery of new pathogens of zebrafish that could potentially be applied to other fish species as well as other models. The identification of a new native pathogen of zebrafish would also be potentially useful for modeling of viral infection within zebrafish. These findings will be of interest to the zebrafish community and the fish immunology community and may also be of interest to immunologists in other model organisms. However, the central finding of the manuscript - the identification and experimental infection with RMBV requires some additional validation.

Concerns:

For the experiments cohousing RMBV-infected pet store zebrafish with laboratory zebrafish, 3 zebrafish are found to have hemorrhage, it is unclear what the status of the other 7 cohoused zebrafish is. Are they also infected, but asymptomatic or are they uninfected? This should be tested for since the presence of asymptomatic carriage in their laboratory populations may change their interpretations, or if it is already known it should be described in the text.

RMBV is described as asymptomatic in pet store fish. However, one possible explanation is that symptomatic RMBV-infected fish had already died prior to purchase or due to disease, these animals weren't made available for purchase so instead only asymptomatic carriers were available. Is there any health data available from the pet store to support completely asymptomatic carriers? This possibility should be discussed, particularly if a percentage of laboratory zebrafish are also found to asymptomatically carry RMBV.

Reviewer #3: In the article "Microbe transmission from pet shop to lab-reared zebrafish reveals a pathogenic birnavirus", the authors describe a novel infectious pathogen that is transmissable between pet-trade zebrafish and lab-reared zebrafish. Their methods are clear and their conclusions are supported by their data. This is an appropriate level of mechanism for a Discovery Report. It leads me to ask many additional questions and get excited for the future of these sorts of analyses, as well as the data they have collected from this initial investigation. There are a few comments and questions that I would like to see addressed: 

1) The authors use the terms metagenomic and metatranscriptomic fairly interchangeably, but it seems that the main work was done using materials derived from RNA extraction. Calling the experimental approach "metagenomics" made initial interpretation of Figure 1D perplexing. 

2) The sentences leading up to the sentence "These results underscore the use of zebrafish rom the pet trade as potential sources of viruses and parasites that are not commonly found in laboratory zebrafish" led me to a different conclusion. If we cannot distinguish between the lab/pet trade sources and there are similar bacterial communities in zebrafish found in labs and pet stores, then how is the pet trade a potential source of novel bacterial infections? I think there might just be a poorly worded sentence or the data from S4 leads us to believe that lab and pet stores have indistinguishable infectious bacterial agents. If the emphasis is on bacterial infections being similar compared to viral infections being unique, then that needs to be more explicitly stated. 

3) In the co-housing experiments, were the sexes of the fish taken into consideration? Was there any instance of fighting among the co-housed? 

4) This may be beyond the scope of the work in this article, but are there different transcriptional responses in infected pet-trade zebrafish compared to the infected laboratory zebrafish? This would be additional support for the claims in the conclusion that cite the importance of "specific pairing of host and virus genomes" in the onset of disease. Given that the pet trade zebrafish were asymptomatic, do the authors believe that they had reduced inflammatory responses OR that they were survivors of a past infection and like IPNV, they are still able to be carriers? 

5) Could the authors provide precise numbers for the symptomatic laboratory zebrafish in the co-housing experiments? What proportion developed lesions? What proportion had identifiable viral RNA in them? 

Overall, this is a well-written and clear article. The authors do not oversell their claims and provide data-based support for nearly all of their conclusions. I look forward to seeing more zebrafish studies on host-RMBV interactions.

---

## [Decision Letter · Decision Letter 2]

27 Mar 2024

Dear Keir,

Thank you for your patience while we considered your revised manuscript "Microbe transmission from pet shop to lab-reared zebrafish reveals a pathogenic birnavirus" for publication as a Discovery Report at PLOS Biology. This revised version of your manuscript has been evaluated by the PLOS Biology editors, the Academic Editor Ken Cadwell and the original reviewers, all of whom support publication. 

I am pleased to say that we can in principle accept your manuscript for publication, provided you address the data and code policy-related requests I detail below. In addition, other formatting requests will be detailed in an email you should receive within 2-3 business days from our colleagues in the journal operations team; no action is required from you until then. Please note that we will not be able to formally accept your manuscript and schedule it for publication until you have completed any requested changes.

Please address the following data and code policy-related requests:

1) DATA POLICY:

a) Supplementary files (e.g., excel). Please ensure that all data files are uploaded as 'Supporting Information' and are invariably referred to (in the manuscript, figure legends, and the Description field when uploading your files) using the following format verbatim: S1 Data, S2 Data, etc. Multiple panels of a single or even several figures can be included as multiple sheets in one excel file that is saved using exactly the following convention: S1_Data.xlsx (using an underscore).

b) Deposition in a publicly available repository. Please also provide the accession code or a reviewer link so that we may view your data before publication. 

**Regardless of the method selected, please ensure that you provide the individual numerical values that underlie the summary data displayed in all the main and supplementary figures, as they are essential for readers to assess your analysis and to reproduce it.**

**Please also ensure that figure legends in your manuscript include information on where the underlying data can be specifically found, and ensure your supplemental data file/s has a legend if possible, or a sufficiently explicit file name to understand what they contain.**

2) CODE POLICY

Per journal policy, if you have generated any custom code during the curse of this investigation, please make it available without restrictions upon publication. Please ensure that the code is sufficiently well documented and reusable, and that your Data Statement in the Editorial Manager submission system accurately describes where your code can be found. 

*Published Peer Review History*

Please note that you will have the opportunity to make the peer review history publicly available, which we strongly encourage. The record will include editor decision letters (with reviews) and your responses to reviewer comments. We will contact you to opt in or out. Please see here for more details:

*Press*

Please do not hesitate to contact me should you have any questions. Thank you again for choosing PLOS Biology for publication and supporting Open Access publishing. We look forward to publishing your study. 

With best wishes,

Nonia

Nonia Pariente, PhD, 

Editor-in-Chief

npariente@plos.org

PLOS Biology

Reviewer remarks:

Reviewer #1: Consistent with comments from all 3 reviewers, this is a super Discovery Report. Thank you to authors for considering all comments and I look forward to following the impact this manuscript will have, as well as future work from this team using the zebrafish infection model. 

Reviewer #2: In the revision, authors have added a new experiment and new data analysis to further establish the level of transmissibility of RMBV and its potential role in activation of interferon responses along with additional experimental details and discussion that address the concerns with the manuscript. The revised manuscript helps further round out the description of RMBV and illustrate this really interesting approach and discovery for the field.

Reviewer #3: The authors have included a new experiment and described the details of it in a way that addresses my major concerns. The revised manuscript clarifies misunderstandings and is improved. My only remaining concern is that there is a supplementary figure 1, 2, and then 5. Is there a distinction between supplementary figures and supplemental files? I continue to think that this is an interesting discovery article and of interest to the zebrafish community.